# Computational Investigation Identifies mTOR as a Primary Binding Target of Medicarpin in Cholangiocarcinoma: Insights from Network Pharmacology and Molecular Docking

**DOI:** 10.3390/life15121828

**Published:** 2025-11-28

**Authors:** Sirinya Sitthirak, Aman Tedasen, Yanisa Rattanapan, Thitinat Duangchan, Hasaya Dokduang, Nawanwat C. Pattaranggoon, Krittamate Saisuwan, Watcharin Loilome, Nisana Namwat

**Affiliations:** 1Department of Medical Technology, School of Allied Health Sciences, Walailak University, Nakhon Si Thammarat 80160, Thailand; sirinya.sit@wu.ac.th (S.S.); aman.te@wu.ac.th (A.T.); yanisa.rt@wu.ac.th (Y.R.); thitinat.du@wu.ac.th (T.D.); 2Research Excellence Center for Innovation and Health Products (RECIHP), Walailak University, Nakhon Si Thammarat 80160, Thailand; 3Hematology and Transfusion Science Research Center, Walailak University, Nakhon Si Thammarat 80160, Thailand; 4Faculty of Medicine, Mahasarakham University, Mahasarakham 44000, Thailand; hasaya.d@msu.ac.th; 5Cholangiocarcinoma Research Institute, Khon Kaen University, Khon Kaen 40002, Thailand; watclo@kku.ac.th; 6Faculty of Medical Technology, Rangsit University, Muang Pathumthani, Pathumthani 12000, Thailand; na-wanwat.p@rsu.ac.th; 7Department of Immunopharmacology, Graduate School of Medicine, Kyoto University, Kyoto 606-8507, Japan; saisuwan.krittamate.2y@kyoto-u.ac.jp; 8Department of Systems Biosciences and Computational Medicine, Faculty of Medicine, Khon Kaen University, Khon Kaen 40002, Thailand

**Keywords:** medicarpin, cholangiocarcinoma, ADME, network pharmacology, molecular docking, signaling pathways

## Abstract

Background: Cholangiocarcinoma (CCA) is an aggressive cancer of biliary tract with poor prognosis and limited therapeutic alternatives. While targeted medicines only benefit a small subset of patients with specific genetic modifications, conventional chemotherapy offers negligible survival advantages. There is an urgent need for novel medicines with multi-target action to combat the diverse and treatment-resistant characteristics of CCA. Methods: An integrative computational strategy combining drug-likeness evaluation, target prediction, network pharmacology, Gene Ontology (GO) and Kyoto Encyclopaedia of Genes and Genomes (KEGG) enrichment analyses, and molecular docking was employed to elucidate the pharmacological profile of medicarpin, a natural pterocarpan derived from *Dalbergia* species. Overlapping targets between medicarpin and CCA-related genes were analysed to construct a protein–protein interaction (PPI) network and identify hub genes. Results: Forty-four overlapping targets were identified, with mTOR, SRC, PIK3CA, and CCND1 emerging as central nodes within the network. Enrichment analyses revealed significant involvement in carcinogenic pathways, including PI3K–Akt/mTOR, ErbB signalling, apoptosis regulation, and drug resistance. Molecular docking demonstrated a stable binding of medicarpin within the catalytic pocket of mTOR (binding energy −9.6 kcal/mol), supported by multiple hydrogen bonds and hydrophobic interactions with key residues essential for kinase activation. Conclusions: This study provides systems-level evidence that medicarpin exerts polypharmacological activity against CCA, with mTOR indicated as a possible mechanistic hub. These findings highlight medicarpin’s potential as a promising multi-target therapeutic candidate and underscore the value of natural compounds in expanding treatment strategies for cholangiocarcinoma.

## 1. Introduction

Cholangiocarcinoma (CCA) is an aggressive cancer of the biliary tract that poses a significant health challenge in Southeast Asia, especially in Northeast Thailand, where liver fluke infection is prevalent [1]. Although surgical techniques and treatment have advanced, the prognosis for people with CCA remains bleak, with five-year survival rates rarely exceeding 10% [2]. Present systemic medicines, such as gemcitabine-cisplatin, yield very marginal advantages, whereas targeted therapies are restricted to a narrow group of patients possessing certain mutations, such as FGFR2 fusions or IDH1 mutations [3,4]. Consequently, it is imperative to discover innovative therapeutic drugs that can target various signalling pathways involved in the growth, invasion, and resistance of CCA [5,6]. Recent advancements in targeted therapy have underscored certain molecular inhibitors with therapeutic promise in CCA. FGFR2 inhibitors (pemigatinib, futibatinib) [7] and IDH1 inhibitors (ivosidenib) [8] have shown clinical efficacy in genetically defined patient cohorts, whereas broader-spectrum inhibitors targeting mTOR, PI3K/AKT, MAPK, and STAT3 pathways have been investigated in preclinical studies with differing levels of effectiveness. Nevertheless, the majority of these drugs encounter constraints such as acquired resistance, route redundancy, and dose limitations due to toxicity. Furthermore, despite the pivotal involvement of various signalling cascades in CCA progression, there are currently no licensed small-molecule inhibitors that simultaneously target several oncogenic networks, highlighting the necessity for multi-target drugs capable of modulating interconnected signalling pathways. Therefore, it is essential to identify novel therapeutic agents that can target many signalling pathways implicated in the proliferation, invasion, and resistance of CCA.

Natural products have historically provided a substantial source of anticancer agents, exhibiting multi-target efficacy and advantageous safety profiles [9,10,11,12,13]. Medicarpin, a natural pterocarpan derived from *Dalbergia* species, has surfaced as a bioactive phytochemical exhibiting several pharmacological properties [14,15]. Prior research has shown its capacity to trigger apoptosis, regulate oxidative stress, and inhibit PI3K/AKT and MAPK signals in many cancer types [9,14,16]. Nonetheless, its therapeutic efficacy in cholangiocarcinoma has not been carefully explored.

The intricacy of CCA, marked by significant intratumoral heterogeneity and disrupted oncogenic networks, necessitates systems-level methodologies for drug discovery [17,18,19]. Network pharmacology integrates chemoinformatics, target prediction, and pathway enrichment to delineate compound-target-disease connections, therefore elucidating the polypharmacological characteristics of natural substances [20,21,22]. This method, when integrated with molecular docking, offers mechanistic insights into drug–protein interactions and prioritises promising targets for subsequent validation [23,24,25].

We utilised a comprehensive computational method to clarify the possible processes of medicarpin in relation to CCA. Through the prediction and intersection of medicarpin-associated targets with CCA-related genes, the construction of protein–protein interaction networks, and the execution of pathway enrichment and molecular docking, we underscore critical oncogenic nodes, namely EGFR, mTOR, STAT3, and CCND1, that may facilitate the anti-CCA effects of medicarpin. This study reveals new mechanistic insights into the pharmacological effects of medicarpin and establishes a logical foundation for its advancement as a potential treatment agent for cholangiocarcinoma.

## 2. Materials and Methods

### 2.1. Chemoinformatics, Drug Likeness, and ADME Prediction

The cheminformatics data and drug-likeness of medicarpin were evaluated using the SwissADME server (http://www.swissadme.ch; accessed on 31 August 2025) [26], an online program specifically designed to estimate pharmacokinetic characteristics, oral bioavailability, and drug-likeness. To assess the drug-likeness of these compounds, Lipinski’s Rule of Five (RO5) was used as a screening criterion for prospective oral pharmaceuticals in humans. The metrics considered comprised molecular weight (MW), lipophilicity (logP), topological polar surface area (TPSA), the count of rotatable bonds, hydrogen bond acceptor (HBA), and hydrogen bond donor (HBD) quantities, in addition to water solubility. The pharmacokinetic characteristics and toxicity (ADMET) profile of medicarpin were systematically evaluated using two computational platforms: pKCSM (https://biosig.lab.uq.edu.au/pkcsm/; accessed on 31 August 2025), which predicts absorption, distribution, metabolism, excretion, and toxicity parameters based on graph-based signatures, and ProTox-III (https://tox.charite.de/protox3/; accessed on 31 August 2025), an advanced web server designed to estimate various toxicity endpoints, including LD50 values, organ-specific toxicities, and potential adverse effects. A detailed flowchart depicting the entire study process is presented in Figure 1.

### 2.2. Prediction of Target Proteins

The identification of targets for the bioactive chemicals produced from medicarpin was conducted using the Swiss Target Prediction databases (http://www.swisstargetprediction.ch/; accessed on 31 August 2025). To achieve this, the canonical SMILES notation of medicarpin was compound and input into the Swiss Target Prediction database. Thereafter, potential targets with elevated probability scores were chosen and subsequently standardised via the UniProt database (http://www.uniprot.org/).

### 2.3. Potential Targets Associated with Cholangiocarcinoma

We used the Human Gene Database (GeneCards, https://www.genecards.org/) [26] to examine and gather the targets relevant to CCA. Utilising the specified search phrase, we identified and aggregated the pertinent targets. Subsequently, we superimposed the projected targets of the chemicals produced from medicarpin with those linked to CCA, culminating in the formation of a Venn diagram. This Venn diagram was created using an online tool available at https://bioinformatics.psb.ugent.be/webtools/Venn/; accessed on 31 August 2025 and visually represents the overlap of identified targets between the therapeutic compounds and the disease. By identifying the shared targets inside this intersection, we derived the target list of drugs from medicarpin, which shows potential for the therapy of CCA.

### 2.4. Gene Ontology and Kyoto Encyclopaedia of Genes and Genomes Pathway Enrichment Analysis

To enhance our comprehension of the significance of important target genes, we performed Gene Ontology (GO) and Kyoto Encyclopaedia of Genes and Genomes (KEGG) pathway enrichment studies [27]. The analyses were conducted utilising ShinyGO 0.82 (http://bioinformatics.sdstate.edu/go/; accessed on 31 August 2025) [28], a bioinformatics tool specifically developed for gene function description and annotation. We employed a significance level of *p* < 0.05 in the enrichment analysis. The outcomes were proficiently illustrated using bubble and bar charts. GO works as an extensive repository for functional genomics, providing precise definitions of gene functions, encompassing molecular functions. Conversely, KEGG includes graphic representations of metabolic pathways and prospective signalling routes. These analyses reveal the functional roles of critical target genes and highlight crucial pathways associated with the chemicals under investigation.

### 2.5. Protein–Protein Interaction (PPI) Network Construction

To investigate the functional associations between the common targets of medicarpin and CCA, a PPI network was generated using STRING v12.0 (species: *Homo sapiens*) with a confidence score threshold of 0.40, thereby retaining medium-confidence interactions (http://string-db.org/). The resulting dataset, comprising proteins represented as nodes and predicted interactions as edges, was exported and subsequently imported into Cytoscape v3.10.1 (https://cytoscape.org/) for visualization and topological analysis. Key network parameters including node degree, clustering coefficient, and betweenness centrality were quantified using the Network Analyzer tool to characterize the overall structural organization. To further identify critical regulatory proteins, CytoHubba (v0.1) was applied to rank nodes based on degree centrality. The top-ten ranked proteins were designated as hub targets, representing potential key regulators for subsequent experimental validation.

### 2.6. Molecular Docking Studies Involving Medicarpin and Hub Genes

The three-dimensional structures of the medicarpin substances (PubChem CID 336327) were obtained from the PubChem database (http://pubchem.ncbi.nlm.nih.gov/). Ligands were optimized by designating bond orderings, angles, and topologies while incorporating any absent and polar hydrogen at pH 7.4. To prepare the ligands for docking, we conducted energy minimization utilizing conjugate steepest descent methods and executed charge addition for ionization correction under the AMBER force field with AM1-BCC charge assignment. The optimization procedures were performed by UCSF Chimaera version 1.17.3. We acquired the three-dimensional crystal structures of the hub target proteins from the Protein Data Bank (PDB, https://www.rcsb.org/). We removed water molecules and small molecule ligands from the protein architecture using BIOVIA Discovery Studio. Subsequently, we used AutoDock technologies to process the hub targets, encompassing stages such as hydrogenation, charge distribution, and atomic type incorporation. The ligand and proteins were converted into AutoDock-compatible formats (PDB and PDBQT).

Molecular docking was conducted via AutoDock version 4.2. The Lamarckian genetic strategy was used for the molecular docking experiment, executed using AutoDock4 software. In the approach, the protein structure was regarded as a stiff molecule, but the ligand was deemed flexible. All other parameters utilized the default parameter values in AutoDockTools (ADT). Fifty genetic algorithm (GA) runs were conducted for conformational sampling, with a population size of 200. A docking box was created to fully enclose the receptor protein’s binding site. The binding energy magnitude was employed to evaluate the probability of an interaction between the receptor and the ligand. The optimum conformation was identified as having minimal binding energy (kcal/mol). The interactions of natural ligands or pharmaceuticals were contrasted with the optimal docked conformation of medicarpin. Binding energy data from the molecular docking procedure was obtained using AutoDockTools 1.5.7. Ultimately, putative protein–ligand interactions and binding modalities were examined and visualized using the BIOVIA Discovery Studio Visualizers program (Accelrys, San Diego, CA, USA) [29].

### 2.7. Molecular Docking (MD) Simulation

MD simulations were performed to investigate the time-dependent behavior of the MTOR–medicarpin complexes, providing insights into how the ligands influence protein flexibility and binding stability. Each complex was prepared at pH 7.0 using the Protein Preparation Wizard, which involved adding hydrogen, assigning bond orders, re-building missing side chains and loops, optimizing hydrogen-bond networks, and sampling water orientations. The systems were then solvated in an orthorhombic box (10 Å × 10 Å × 10 Å) filled with TIP3P water molecules, neutralized with Na^+^ and Cl^−^ ions to a concentration of 0.15 M and parameterized for simulation. Production runs were conducted for 300 ns under an NPT ensemble at 310 K and 1.01 bar. Long-range electrostatics were calculated using the Smooth Particle Mesh Ewald (PME) method, while the solvent was modeled with a simple point-charge representation. Trajectory analyses, performed via the Simulation Interaction Diagram wizard, included ligand–protein contact maps, root-mean-square deviation (RMSD) profiles, root-mean-square fluctuation (RMSF) plots, and timeline interaction analyses for both ligand and protein atoms. All simulations and subsequent analyses were carried out using Desmond (Schrödinger), yielding a comprehensive view of structural stability, conformational dynamics, and key interaction hotspots throughout the simulation period [30].

### 2.8. Survival Analysis of mTOR Expression in Cholangiocarcinoma Patients

We conducted a survival analysis utilising The Cancer Genome Atlas cholangiocarcinoma cohort (TCGA-CHOL) to examine the impact of mTOR expression on the overall survival (OS) of patients. Gene expression and clinical survival data were sourced from the Gene Expression Profiling Interactive study 2 (GEPIA2; http://gepia2.cancer-pku.cn/; accessed on 7 September 2025), a prominent platform for integrated cancer genomics study. The mTOR expression levels in CCA patients were categorised into two groups: high expression and low expression, determined by the median cut-off value. A Kaplan–Meier survival curve was later constructed to compare the overall survival results of these two groups. The prognostic relevance of mTOR was assessed by the hazard ratio (HR) with 95% confidence intervals, and the log-rank *p*-value was computed to establish statistical significance. This approach allowed us to ascertain if increased mTOR expression is associated with unfavourable survival outcomes in CCA patients, offering further understanding of its potential function as a prognostic biomarker.

## 3. Results

### 3.1. Procedure for Network Pharmacology Assessment of Medicarpin in Relation to Cholangiocarcinoma

The comprehensive workflow utilised in this study is encapsulated in Figure 1. Potential medicarpin targets were initially identified using the SwissTarget database, while cholangiocarcinoma-associated genes were obtained from GeneCards. Candidate targets were identified as overlapping genes between medicarpin targets and CCA-associated genes. The identified targets underwent network design and functional investigation. A protein–protein interaction (PPI) network was constructed to identify hub genes, followed by functional enrichment analysis, encompassing Gene Ontology (GO) and Kyoto Encyclopaedia of Genes and Genomes (KEGG) pathway analyses. Molecular docking was used to validate the anticipated contacts, followed by molecular dynamics (MD) simulation to evaluate the stability of the medicarpin-target protein complexes. Additionally, survival analysis was conducted to assess the prognostic relevance of the discovered hub genes in CCA patients.

### 3.2. Chemoinformatics, Drug-likeness, and ADME-Tox Profiling of Medicarpin

Thorough chemoinformatics and ADME-Tox evaluations were conducted to determine the pharmacological viability of medicarpin as a drug-like compound prior to target identification and docking. The physicochemical profile predicted by SwissADME (Table 1) indicated that medicarpin (C_16_H_14_O_4_; MW = 270.28 g/mol) adheres to Lipinski’s Rule of Five, demonstrating no violations and meeting the criteria of Ghose, Veber, Egan, and Muegge filters. The compound’s moderate lipophilicity (consensus LogP = 2.53) and low topological polar surface area (TPSA = 47.9 Å^2^) suggest an advantageous equilibrium between solubility and permeability, aligning with its potential for passive intestinal absorption and blood–brain barrier (BBB) passage. The anticipated high gastrointestinal absorption and a bioavailability score of 0.55 indicate favourable oral exposure potential. Medicinal chemistry assessments indicated no PAINS or Brenk alarms, and a synthetic accessibility score of 3.54 suggests that laboratory synthesis is viable, facilitating its progression for additional screening.

The ADMET forecast conducted via the pKCSM platform (Table 2) substantiated these findings. Medicarpin exhibited remarkable intestinal absorption (95.2%), satisfactory Caco-2 permeability (log Papp = 1.25), and advantageous skin permeability (log Kp = −2.82). The molecule neither serves as a substrate nor an inhibitor of P-glycoprotein, thereby reducing the possibility of bioavailability loss due to efflux. Distribution metrics indicated little plasma protein binding (fraction unbound = 0.04) and a modest volume of distribution (log VDss = 0.07), aligning with effective tissue penetration. The anticipated blood–brain barrier permeability (log BB = 0.32) and central nervous system permeability (log PS = −1.84) suggest possible central exposure without significant neurotoxicity risk.

In silico metabolism analysis revealed medicarpin as a substrate for CYP3A4 and an inhibitor of CYP1A2, CYP2C19, CYP2C9, CYP2D6, and CYP3A4, indicating potential multi-enzyme interactions that necessitate in vitro validation for drug–drug interaction risk. The overall clearance rate (log mL/min/kg = 0.27) suggested considerable systemic elimination, but renal excretion through OCT2 was anticipated to be non-existent. Toxicological assessments indicated that medicarpin was non-hepatotoxic, non-cardiotoxic (hERG I/II negative), and non-skin-sensitising, with tolerable acute (LD_50_ = 2.51 mol/kg) and chronic (LOAEL = 1.88 log mg/kg/day) toxicity levels.

Complementary Protox-III predictions (Table 3) offered comprehensive toxicity endpoints. Medicarpin is categorised as Class IV (LD_50_ ≈ 1000 mg/kg, “harmful if ingested”), which is characteristic of orally accessible phytochemicals. Hepatotoxicity and pulmonary toxicity were identified as active organ toxicities, with probabilities of 0.69 and 0.98, respectively, while nephrotoxicity and cardiotoxicity were deemed inactive, with probabilities equal to or exceeding 0.77. The chemical was forecasted to be inactive for mutagenicity and carcinogenicity (probabilities > 0.9) and showed no activity against principal stress-response pathways (Nrf2/ARE, p53, HSE). Medicarpin demonstrated a significant interaction with aromatase and oestrogen receptor-α (ERα) with a high likelihood (about 1.0), indicating its phyto-oestrogenic structure, which aligns with previous biochemical findings on selective oestrogen-receptor regulation.

The combined data on drug-likeness, pharmacokinetics, and toxicity indicate that medicarpin has a favourable oral bioavailability, minimal systemic toxicity, and adequate synthetic accessibility, providing a solid basis for future network pharmacology and molecular docking studies to identify its anti-cholangiocarcinoma targets.

### 3.3. Identification of Targets and Analysis of Networks

To clarify the therapeutic significance of medicarpin in CCA, possible targets were discovered via SwissTargetPrediction, and CCA-related genes were obtained from the GeneCards database. A total of 130 targets were identified for medicarpin, and 2506 genes were linked to CCA. A comparative study identified 44 overlapping targets (Figure 2A), which serve as potential mediators of medicarpin’s efficacy against CCA.

A protein–protein interaction (PPI) network was established using the STRING database to investigate the functional connections among these targets (Figure 2B). The resultant network displayed dense linkages, emphasising crucial signalling nodes potentially affected by medicarpin. Topological analysis with the cytoHubba plug-in in Cytoscape was employed to discover hub genes based on degree scores. The primary ten hub targets comprised SRC, mTOR, ESR1, CASP3, PIK3CA, CCND1, GSK3B, PARP1, KIT, and KDR (Figure 2C).

SRC and MTOR appeared as central nodes with the greatest degree values, indicating their crucial significance in the pharmacological mechanism of medicarpin against CCA. These hub targets are essential regulators of carcinogenic pathways encompassing cell survival, death, and proliferation; therefore, they represent prospective molecular targets for further validation.

### 3.4. Gene Ontology Enrichment Analysis

A complete Gene Ontology (GO) enrichment analysis was conducted using ShinyGO to further elucidate the biological implications of the 44 overlapped targets of medicarpin and CCA. A total of 1842 Gene Ontology (GO) terms were highly enriched (FDR < 0.05), including 983 biological process (BP) terms, 438 cellular component (CC) terms, and 421 molecular function (MF) items.

In the biological process category (Figure 3A), the most enriched phrases pertained to protein phosphorylation, regulation of kinase activity, modulation of cell death, apoptotic processes, and cellular responses to chemical stimuli. The findings indicate that medicarpin may influence its therapeutic effects by modulating signalling pathways that regulate survival and apoptosis. In the cellular component category (Figure 3B), notable enrichment was detected for phosphatidylinositol 3-kinase complexes, chromosomal regions, nuclear specks, membrane rafts, and receptor complexes, underscoring the participation of essential subcellular compartments pertinent to signal transduction and transcriptional regulation. In the molecular function category (Figure 3C), the most enriched phrases encompassed protein tyrosine kinase activity, protein serine/threonine kinase activity, phosphotransferase activity, ATP binding, and nucleotide binding, demonstrating the fundamental importance of kinases as significant druggable targets in CCA. These data suggest that medicarpin may influence kinase-dependent oncogenic signalling pathways and apoptotic processes in CCA, underscoring its potential as a multi-target therapeutic drug.

### 3.5. KEGG Pathway Enrichment Analysis

KEGG pathway enrichment analysis was conducted on the 44 overlapped targets to identify the signalling cascades likely regulated by medicarpin in CCA. A total of 156 pathways with substantial enrichment (FDR < 0.05) were found. The highest-ranked pathways were primarily linked to cancer and oncogenic signalling (Figure 4). The topics encompassed central carbon metabolism in cancer, non-small-cell lung cancer, the prolactin signalling pathway, resistance to EGFR tyrosine kinase inhibitors, melanoma, chronic myeloid leukaemia, prostate cancer, ErbB signaling pathway, endocrine resistance, colorectal cancer, the thyroid hormone signalling pathway, and breast cancer. Several CCA-related signalling pathways, including PI3K-Akt signalling, the ErbB signalling, and proteoglycans in cancer, were significantly enriched. The results underscore medicarpin’s extensive role in regulating cancer-related signalling, especially pathways that control cell proliferation, metabolism, apoptosis, and treatment resistance. The enhancement of PI3K-Akt and EGFR signalling pathways, recognised as key contributors to CCA progression, further emphasises the potential of medicarpin as a multi-target treatment candidate for this cancer.

### 3.6. Mapping Pathways of Intersecting Targets

To obtain a more profound understanding of the functional significance of medicarpin-associated targets in cholangiocarcinoma, the 44 overlapped genes were mapped onto the KEGG “Pathways in Cancer” module utilising Pathview. Figure 5 illustrates that several nodes associated with medicarpin targets were allocated among various hallmark oncogenic pathways, including PI3K-Akt, MAPK, mTOR, Wnt, apoptosis, and cell cycle signalling cascades. Central regulators, including PIK3CA, mTOR, SRC, CCND1, ESR1, and CASP3, were significantly enriched in essential modules that regulate proliferation, apoptosis evasion, and therapeutic resistance. The presence of these genes at critical signalling nodes indicates that medicarpin may have multi-targeted effects by influencing interconnected oncogenic networks instead of operating on a singular pathway. This pathway-level visualisation emphasises the polypharmacological potential of medicarpin in addressing CCA development and identifies key axes, especially the PI3K-Akt/mTOR signalling cascade, as viable treatment avenues for future validation.

### 3.7. Validation of Hub Targets via Molecular Docking

Molecular docking was conducted to validate the interaction between medicarpin and the principal hub targets identified from the PPI network. mTOR was chosen for structural docking research because of its pivotal involvement in cancer development and significant enrichment in KEGG pathways among the hub proteins. Table 4 illustrates the extensive docking results for all ten potential proteins, identifying mTOR as one of the most robust binding targets of medicarpin due to its very favourable binding energy and interaction profile.

Figure 6 illustrates that medicarpin was securely situated within the catalytic binding pocket of mTOR, exhibiting a binding energy of −9.6 kcal/mol (a threshold of <−9.0 kcal/mol is deemed stable). The association was reinforced by several hydrogen bonds, particularly with THR887, LYS890, and VAL882, in addition to hydrophobic interactions with ILE881, ILE963, MET953, PHE961, and TYR867. Moreover, π–π and alkyl interactions were detected, significantly augmenting the binding stability. The docking analysis indicates that medicarpin can successfully bind to and perhaps decrease mTOR activity, a pathway often dysregulated in CCA. These data substantiate the concept that mTOR may serve as a principal pharmacological target of medicarpin in CCA, necessitating additional biochemical and cellular confirmation.

### 3.8. Molecular Dynamics Simulation of the Medicarpin–mTOR Complex

A 300-nanosecond molecular dynamics simulation was conducted to assess the stability and dynamic behaviour of medicarpin within the mTOR active site. The root mean square deviation (RMSD) plot (Figure 6A) indicated that the mTOR backbone attained structural equilibrium after roughly 60 ns, sustaining a steady variation between 2.5 and 3.5 Å for the duration of the journey. The ligand RMSD closely mirrored the protein trajectory, demonstrating that medicarpin remained securely situated within the binding pocket without notable positional deviation or separation. The little variation of the Cα atoms verified that the system was adequately equilibrated throughout the simulation.

The root mean square fluctuation (RMSF) profile (Figure 6B) indicated restricted flexibility in the majority of residues, with slight changes noted in loop areas far from the binding site. The residues constituting the catalytic domain of mTOR displayed negligible variation, indicating that medicarpin binding did not disrupt the overall structure of the protein.

The protein–ligand interaction analysis additionally corroborated the stability of the combination. Interaction fraction plots (Figure 6C) demonstrated that medicarpin sustained enduring hydrogen bonds and hydrophobic interactions with essential active-site residues, notably Ala805, Ser809, Val882, and Met953, which are recognised for their pivotal roles in substrate recognition. The timeline depiction of protein–ligand interactions (Figure 6D) demonstrated persistent contact during the 300 ns duration, underscoring the robust and stable binding affinity of medicarpin to mTOR.

The 2D ligand-interaction diagram (Figure 6E) demonstrated that medicarpin establishes many hydrogen bonds with Ala805, Ser809, and Trp812, in addition to hydrophobic interactions with Val882, Tyr867, and Met953. These contacts are situated near the ATP-binding cleft, a region critical for kinase activation.

The MD data collectively imply that the medicarpin–mTOR complex is structurally stable and energetically advantageous. The enduring presence of hydrogen bonds and hydrophobic interactions in the catalytic pocket indicates that medicarpin may serve as an effective mTOR inhibitor, stabilising the kinase’s inactive conformation and potentially influencing downstream PI3K/AKT/mTOR signalling pathways associated with CCA progression.

As shown in Table 5, docking analysis of the medicarpin–mTOR complex initially revealed hydrogen bonds with VAL882, THR887, and LYS890, alongside hydrophobic contacts involving ILE831, TYR867, ILE881, MET953, PHE961, and ILE963. However, the MD simulation timeline demonstrated a dynamic reorganization of these interactions over 300 ns. Early hydrogen bonding (50 ns) shifted toward ALA805, ILE881, and THR887, with subsequent stabilization around ALA805, SER806, VAL882, and THR887 at 100–150 ns. By 200–300 ns, additional residues such as ASP950 and GLY970 emerged, indicating an expansion of the hydrogen bond network. Similarly, hydrophobic interactions evolved from the docking profile to include MET804, TRP812, ILE879, and ILE968 during the simulation, while persistent contacts with ILE831, TYR867, ILE881, MET953, and ILE963 were maintained throughout. Notably, PHE961, initially observed in docking, reappeared at 300 ns, suggesting its role in long-term stabilization. Overall, the MD trajectory highlights that medicarpin maintains core interactions while progressively engaging additional residues, reflecting enhanced stability and adaptability of the ligand–protein complex compared to the static docking snapshot.

### 3.9. Prognostic Significance of mTOR Expression in Cholangiocarcinoma Patients

To assess the predictive significance of mTOR expression in CCA, patients were classified into high- and low-expression cohorts (n = 18 each) according to the median expression level. Kaplan–Meier survival analysis revealed a slight survival benefit in the high-mTOR group relative to the low-mTOR group; still, the difference lacked statistical significance (log-rank *p* = 0.34). The hazard ratio (HR) for patients exhibiting elevated mTOR expression was 0.64, suggesting a non-significant tendency towards lower mortality risk (Figure 7). Despite the lack of statistical robustness, the identified survival pattern corroborates the concept that the activation of the PI3K/Akt/mTOR pathway may enhance a more advantageous tumour phenotype in certain patients. These findings necessitate validation in a larger cohort with comprehensive transcriptome and phosphoproteomic analysis to elucidate the prognostic significance of mTOR activation in CCA.

## 4. Discussion

This study presents the initial systems-level evidence that medicarpin, a naturally occurring pterocarpan sourced from Dalbergia species, demonstrates multi-targeted anticancer actions against CCA. Through an integrative computational approach that incorporates network pharmacology, molecular docking, and molecular dynamics (MD) simulation, we identified mTOR as a key mechanistic hub facilitating the pharmacological effects of medicarpin. The convergence of medicarpin-associated targets within PI3K/AKT/mTOR, MAPK, and apoptosis-related signalling pathways highlights their ability to modulate critical oncogenic pathways that control tumour growth, survival, and treatment resistance in CCA [6,17,19].

mTOR functions as a principal regulator of cellular growth and metabolism, often exhibiting overactivation in the CCA because of dysregulated PI3K/AKT signalling, gene amplification, or the absence of upstream inhibitors such as PTEN [4,31]. Continuous mTOR activation facilitates metabolic reprogramming, epithelial–mesenchymal transition (EMT), and chemoresistance [5,32]. Our docking and 300 ns molecular dynamics simulation demonstrated that medicarpin forms stable hydrogen bonds with Ala805, Ser809, and Val882, as well as hydrophobic interactions with Met953 and Tyr867, while exhibiting a low root mean square deviation and a constrained root mean square fluctuation profile throughout the trajectory. The observations suggest a robust and energetically advantageous binding that may allosterically stabilise mTOR in its inactive conformation, thus suppressing downstream PI3K/AKT signalling. These computational findings offer mechanistic insights but should be taken with caution, as in silico stability does not necessarily correlate with functional inhibition in biological systems. Subsequent research should compare medicarpin with structurally analogous flavonoids and assess its efficacy across genetically varied CCA subtypes to ascertain subgroup-specific responses.

The physicochemical and pharmacokinetic properties of medicarpin further substantiate its drug-like nature. It adheres to the criteria established by Lipinski, Ghose, Veber, and Egan, demonstrating elevated gastrointestinal absorption, favourable bioavailability, and an absence of hepatotoxic or cardiotoxic risks [33,34,35]. Medicarpin’s anticipated safety and oral bioavailability set it apart from several synthetic kinase inhibitors that experience dose-limiting toxicities or inadequate metabolic stability [36]. The polypharmacological properties of medicarpin may provide a therapeutic benefit for CCA, a cancer distinguished by significant genomic heterogeneity and compensatory signalling network reconfiguration [18,37,38,39].

Prior research indicates that medicarpin promotes apoptosis and enhances tumour cell sensitivity to TRAIL-mediated cytotoxicity via the ROS-JNK-CHOP pathway in leukaemia [16], activates the PI3K/AKT/FoxO pathway to safeguard endothelial cells under stress [14], and inhibits cancer cell proliferation through MAPK suppression [9]. Our study expands these anticancer methods to CCA, indicating that medicarpin’s multifaceted activities converge on the mTOR signalling pathway. This pathway convergence reflects findings in other natural chemicals, such as resveratrol and curcumin, which influence interrelated oncogenic cascades.

The merging of network pharmacology with molecular dynamics simulations represents a cost-effective and mechanistically insightful approach to identify druggable vulnerabilities in complex malignancies like CCA. This in silico model connects computational predictions with experimental validation, facilitating the prioritisation of natural drugs with superior ADME toxicity profiles for subsequent preclinical investigation. Although these encouraging results, many restrictions must be recognised. This study is exclusively computational, and experimental validation to confirm the expected binding or subsequent effects of medicarpin on mTOR signalling has not yet been conducted. Secondly, our research concentrated on a singular ligand, which may not comprehensively represent the wider pharmacological spectrum of natural molecules with analogous scaffolds. Third, network- and docking-based target screening may generate prediction biases due to reliance on database completeness, algorithm limitations, and the structural availability of target proteins. The survival trends obtained from public databases exhibit irregularities that restrict their interpretability and should be considered exploratory rather than definitive. Mitigating these constraints using cellular assays, kinase inhibition investigations, and multi-ligand comparative profiling will be crucial in forthcoming research.

Our findings collectively underscore medicarpin as a promising lead drug with strong binding affinity and persistent inhibition of mTOR, bolstered by advantageous pharmacokinetics and multi-pathway modulation. Subsequent research should corroborate these results via biochemical assays of mTOR phosphorylation (p-mTOR, p70S6K, 4EBP1) and assess the synergistic efficacy of medicarpin in conjunction with FGFR2 or IDH1 inhibitors in CCA cell and xenograft models.

This study reveals a new molecular paradigm via which medicarpin exhibits multi-target anticancer effects via direct mTOR inhibition and comprehensive signaling regulation. The results identify medicarpin as a promising natural scaffold for advanced multi-kinase therapies in cholangiocarcinoma, a condition that necessitates more effective and safer treatment approaches.

## 5. Conclusions

This study offers the inaugural thorough systems-level analysis of medicarpin’s therapeutic efficacy against cholangiocarcinoma. Utilising an integrative in silico methodology that encompasses network pharmacology, molecular docking, and long-timescale molecular dynamics simulations, we identified mTOR as a crucial pharmacological hub, facilitating medicarpin’s multi-target activity. Medicarpin exhibited a stable, high-affinity engagement within the catalytic pocket of mTOR, facilitated by enduring hydrogen bonding and hydrophobic interactions that preserved conformational stability across a 300 ns simulation.

Medicarpin-associated targets were significantly enriched in oncogenic pathways critical to CCA progression, including PI3K/AKT/mTOR, MAPK, apoptosis, and cell-cycle regulation, underscoring its capacity to simultaneously influence multiple signaling pathways that promote tumour proliferation and therapeutic resistance. Its advantageous ADMET and drug-likeness characteristics further emphasize its potential for translation as an orally accessible, low-toxicity natural scaffold for kinase inhibition.

The activation of the PI3K/Akt/mTOR pathway is repeatedly identified as a principal oncogenic driver in CCA, facilitating unregulated cell proliferation, survival, and chemoresistance. Dokduang et al. (2013) found that phosphorylated mTOR and its downstream effectors, p70S6K and 4E-BP1, were significantly increased in CCA tissues and cell lines, showing persistent activation of this pathway and its correlation with tumour aggressiveness and unfavourable prognosis [40]. Inhibition of mTOR signalling via rapamycin substantially curtailed CCA cell proliferation and triggered apoptosis, thereby affirming the pathway’s therapeutic significance. Yothaisong et al. (2013) similarly shown that the combined inhibition of PI3K and mTOR with NVP-BEZ235 significantly decreased cell viability and increased apoptosis in CCA cells, highlighting the cancer’s need on PI3K/Akt/mTOR signalling for survival [41]. These findings offer robust biological validation for our computational results, which designate mTOR as a critical pharmacological target of Medicarpin. The collective evidence indicates that Medicarpin’s anticipated mTOR inhibition may yield significant therapeutic benefits in cholangiocarcinoma by modulating this highly active carcinogenic pathway.

These findings establish medicarpin as a promising candidate for multi-pathway intervention in CCA, providing a molecular basis for further in vitro and in vivo validation. This study not only uncovers a new mTOR-targeted mechanism but also demonstrates how computational network pharmacology can expedite the identification of phytochemical-based treatments for difficult-to-treat malignancies. Future preclinical investigations incorporating phosphoproteomic and metabolomic analysis will be essential to comprehensively elucidate medicarpin’s systemic anticancer mechanism and its therapeutic synergy with current targeted therapies.

## Figures and Tables

**Figure 1 life-15-01828-f001:**
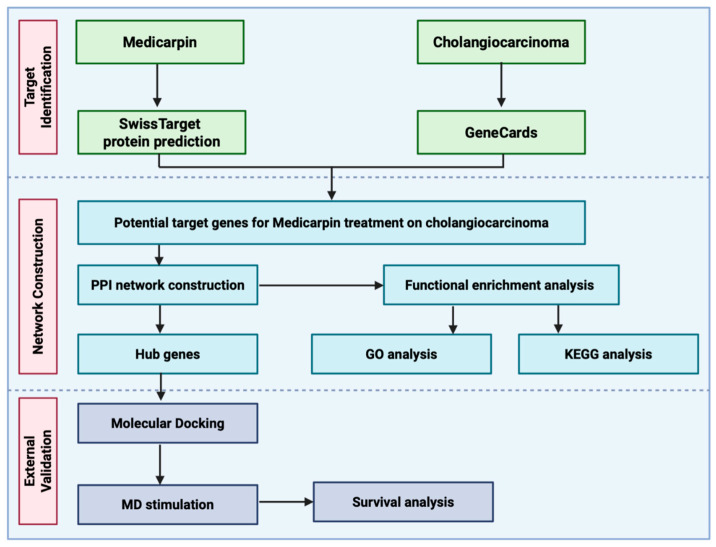
General workflow of network pharmacology and molecular docking studies of current work.

**Figure 2 life-15-01828-f002:**
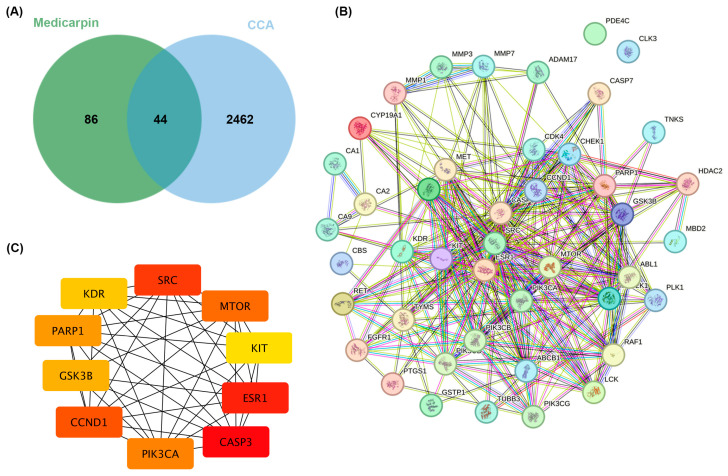
Protein–protein interaction (PPI) network and hub gene analysis. (**A**) A Venn diagram illustrating the intersecting relationship of targets between bioactive chemicals from medicarpin and cholangiocarcinoma. (**B**) The PPI network, executed by the STRING database, comprises 44 shared target networks. (**C**) The PPI network of the ten principal hub genes was analysed using the Cytoscape plugin cytoHubba.

**Figure 3 life-15-01828-f003:**
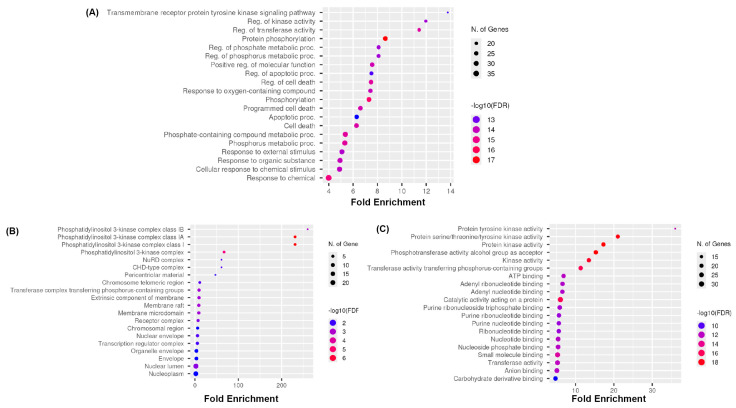
Gene Ontology enrichment analysis for the targets of bioactive chemicals from medicarpin in the treatment of cholangiocarcinoma (*p* value < 0.05). Gene Ontology study of biological processes (**A**), cellular components (**B**), and molecular functions (**C**) of possible target genes of medicarpin in oncology.

**Figure 4 life-15-01828-f004:**
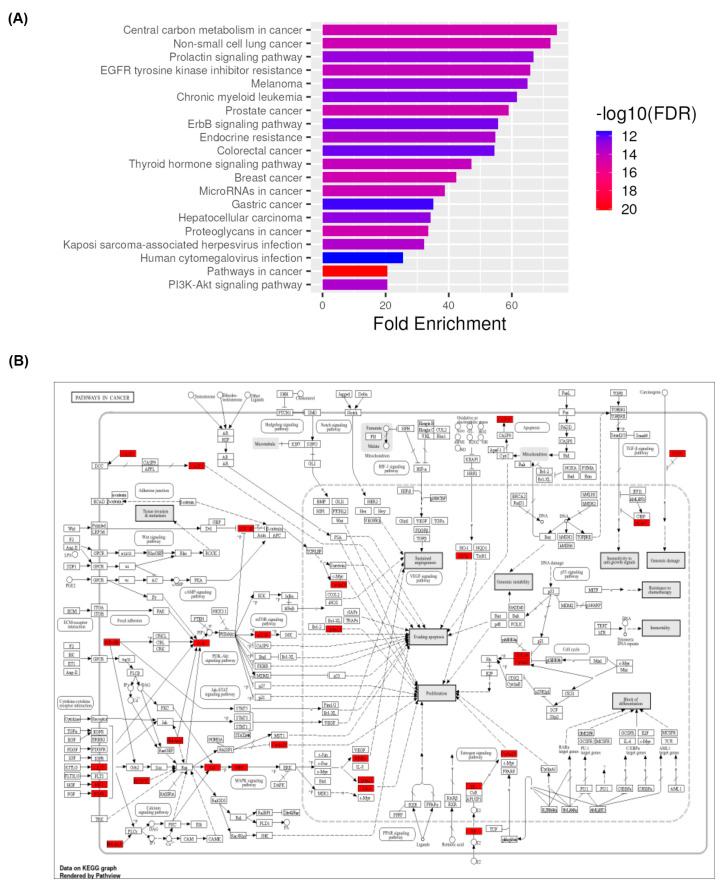
(**A**) KEGG pathway enrichment and cancer-related mapping of targets associated with medicarpin in cholangiocarcinoma. KEGG pathway enrichment analysis of 44 overlapping targets between medicarpin and cholangiocarcinoma, ordered by fold enrichment. The most notably enhanced pathways encompass PI3K-Akt signalling, ErbB signalling, and other oncogenic processes, including those associated with prostate, colorectal, and hepatocellular carcinoma, underscoring medicarpin’s extensive involvement with tumour-promoting networks. Colour intensity denotes statistical significance (–log_10_ FDR). (**B**) Mapping of intersecting targets into the KEGG “Pathways in Cancer” diagram (23). Red boxes denote proteins or genes that may be influenced by medicarpin, allocated within essential oncogenic modules that regulate proliferation, survival, apoptosis, angiogenesis, and metastasis. The prevalence of hits in the PI3K/AKT/mTOR and MAPK pathways highlights medicarpin’s ability to disrupt key signalling centres that promote cholangiocarcinoma advancement.

**Figure 5 life-15-01828-f005:**
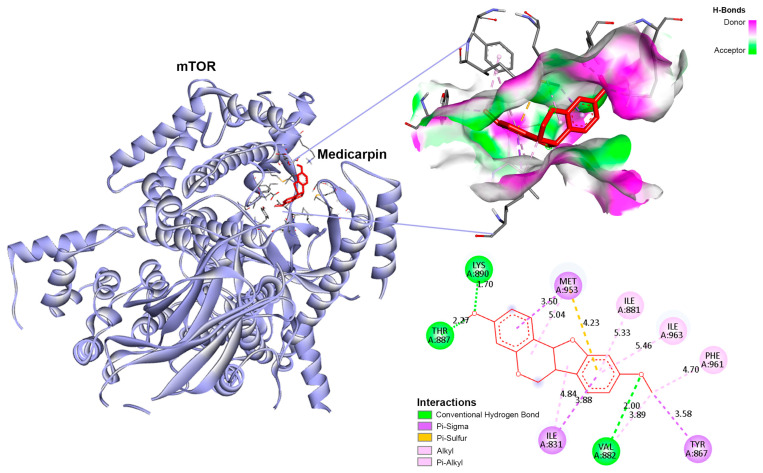
Molecular docking studies of active compounds from medicarpin and against top hub targets.

**Figure 6 life-15-01828-f006:**
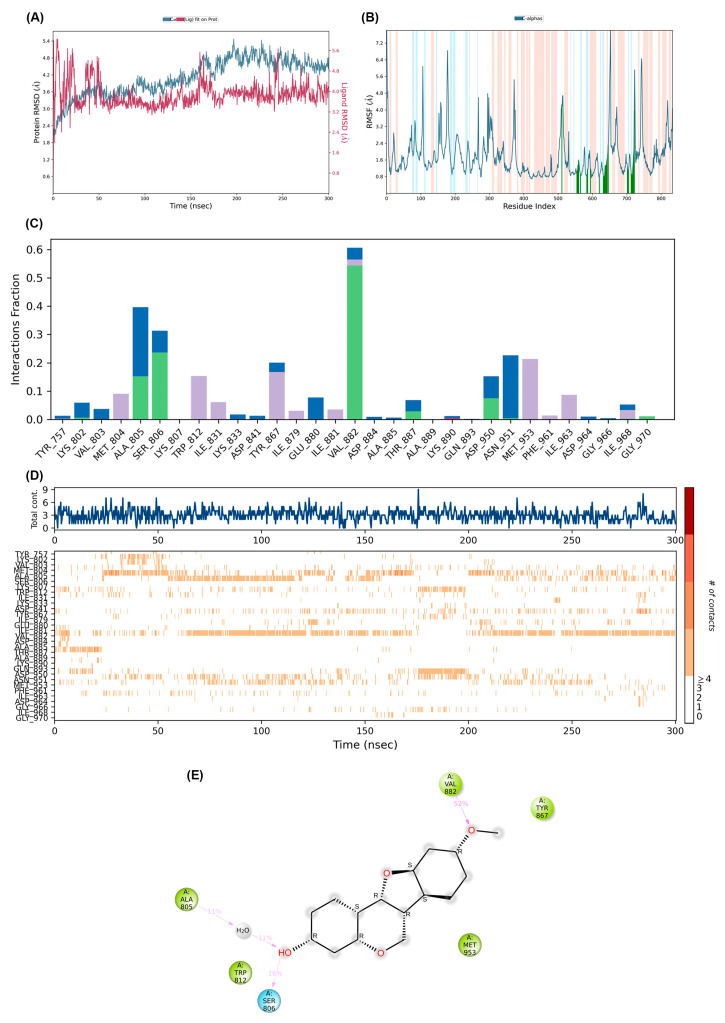
Molecular dynamics simulation of the medicarpin–mTOR complex. (**A**) Root mean square deviation (RMSD) plot of the protein backbone and ligand during a 300 ns simulation. (**B**) Root mean square fluctuation (RMSF) of residues indicating flexibility of the binding region. (**C**) Histogram showing interaction fractions between medicarpin and key residues of mTOR. (**D**) Timeline representation of protein–ligand contacts throughout the simulation period. (**E**) Two-dimensional ligand interaction diagram of medicarpin within the mTOR active site.

**Figure 7 life-15-01828-f007:**
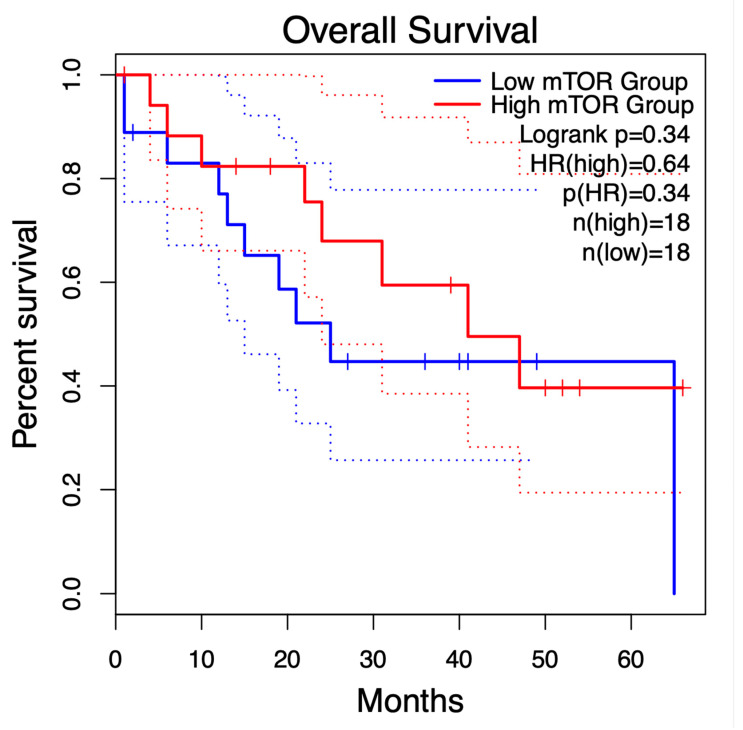
Kaplan–Meier analysis of overall survival based on mTOR expression in cholangiocarcinoma.

**Table 1 life-15-01828-t001:** Evaluation of the physicochemical properties and drug-likeness of Medicarpin using the SwissADME service.

**Properties**	
**Physicochemical Properties**	
Formula	C_16_H_14_O_4_
Molecular weight	270.28 g/mol
Num. heavy atoms	20
Num. arom. heavy atoms	12
Fraction Csp3	0.25
Num. rotatable bonds	1
Num. H-bond acceptors	4
Num. H-bond donors	1
Molar Refractivity	73.17
TPSA	47.92 Å^2^
**Lipophilicity**	
Log Po/w (iLOGP)	2.57
Log Po/w (XLOGP3)	2.77
Log Po/w (WLOGP)	2.69
Log Po/w (MLOGP)	1.87
Log Po/w (SILICOS-IT)	2.75
Consensus Log Po/w	2.53
**Water Solubility**	
Log S (ESOL)	−3.64
Solubility	6.21 × 10^−2^ mg/mL; 2.30 × 10^−4^ mol/L
Class	Soluble
Log S (Ali)	−3.43
Solubility	1.00 × 10^−1^ mg/mL; 3.70 × 10^−4^ mol/L
Class	Soluble
Log S (SILICOS-IT)	−4.31
Solubility	1.32 × 10^−2^ mg/mL; 4.90 × 10^−5^ mol/L
Class	Moderately soluble
**Pharmacokinetics**	
GI absorption	High
BBB permeant	Yes
P-gp substrate	Yes
CYP1A2 inhibitor	Yes
CYP2C19 inhibitor	Yes
CYP2C9 inhibitor	No
CYP2D6 inhibitor	Yes
CYP3A4 inhibitor	Yes
Log Kp (skin permeation)	−5.98 cm/s
**Drug-likeness**	
Lipinski	Yes; 0 violation
Ghose	Yes
Veber	Yes
Egan	Yes
Muegge	Yes
Bioavailability Score	0.55
**Medicinal Chemistry**	
PAINS	0 alert
Brenk	0 alert
Lead-likeness	Yes
Synthetic accessibility	3.54

**Table 2 life-15-01828-t002:** Prediction of medicarpin’s pharmacokinetic (ADMET) properties using pKCSM.

Property	Model Name	Predicted Value	Unit
**Absorption**	Water solubility	−3.459	Numeric (log mol/L)
	Caco2 permeability	1.246	Numeric (log Papp in 10^−6^ cm/s)
	Intestinal absorption (human)	95.188	Numeric (% Absorbed)
	Skin Permeability	−2.819	Numeric (log Kp)
	P-glycoprotein substrate	No	Categorical (Yes/No)
	P-glycoprotein I inhibitor	No	Categorical (Yes/No)
	P-glycoprotein II inhibitor	No	Categorical (Yes/No)
**Distribution**	VDss (human)	0.065	Numeric (log L/kg)
	Fraction unbound (human)	0.04	Numeric (Fu)
	BBB permeability	0.324	Numeric (log BB)
	CNS permeability	−1.838	Numeric (log PS)
**Metabolism**	CYP2D6 substrate	No	Categorical (Yes/No)
	CYP3A4 substrate	Yes	Categorical (Yes/No)
	CYP1A2 inhibitor	Yes	Categorical (Yes/No)
	CYP2C19 inhibitor	Yes	Categorical (Yes/No)
	CYP2C9 inhibitor	Yes	Categorical (Yes/No)
	CYP2D6 inhibitor	No	Categorical (Yes/No)
	CYP3A4 inhibitor	Yes	Categorical (Yes/No)
**Excretion**	Total Clearance	0.273	Numeric (log mL/min/kg)
	Renal OCT2 substrate	No	Categorical (Yes/No)
**Toxicity**	AMES toxicity	Yes	Categorical (Yes/No)
	Max. tolerated dose (human)	−0.102	Numeric (log mg/kg/day)
	hERG I inhibitor	No	Categorical (Yes/No)
	hERG II inhibitor	No	Categorical (Yes/No)
	Oral Rat Acute Toxicity (LD50)	2.512	Numeric (mol/kg)
	Oral Rat Chronic Toxicity (LOAEL)	1.875	Numeric (log mg/kg_bw/day)
	Hepatotoxicity	No	Categorical (Yes/No)
	Skin Sensitisation	No	Categorical (Yes/No)
	T. Pyriformis toxicity	0.688	Numeric (log ug/L)
	Minnow toxicity	0.657	Numeric (log mM)

**Table 3 life-15-01828-t003:** Prediction of medicarpin’s pharmacokinetic and toxicity (ADMET) properties using ProTox-III.

Classification	Target	Prediction	Probability
Organ toxicity	Hepatotoxicity	Active	0.69
Organ toxicity	Neurotoxicity	Active	0.87
Organ toxicity	Nephrotoxicity	Inactive	0.9
Organ toxicity	Respiratory toxicity	Active	0.98
Organ toxicity	Cardiotoxicity	Inactive	0.77
Toxicity end points	Carcinogenicity	Inactive	0.62
Toxicity end points	Immunotoxicity	Active	0.96
Toxicity end points	Mutagenicity	Inactive	0.97
Toxicity end points	Cytotoxicity	Inactive	0.93
Toxicity end points	BBB-barrier	Inactive	1
Toxicity end points	Ecotoxicity	Active	0.73
Toxicity end points	Clinical toxicity	Inactive	0.56
Toxicity end points	Nutritional toxicity	Inactive	0.74
Tox21-Nuclear receptor signalling pathways	Aryl hydrocarbon Receptor (AhR)	Inactive	0.97
Tox21-Nuclear receptor signalling pathways	Androgen Receptor (AR)	Inactive	0.99
Tox21-Nuclear receptor signalling pathways	Androgen Receptor Ligand Binding Domain (AR-LBD)	Inactive	0.99
Tox21-Nuclear receptor signalling pathways	Aromatase	Active	1
Tox21-Nuclear receptor signalling pathways	Estrogen Receptor Alpha (ER)	Active	0.99
Tox21-Nuclear receptor signalling pathways	Estrogen Receptor Ligand Binding Domain (ER-LBD)	Active	1
Tox21-Nuclear receptor signalling pathways	Peroxisome Proliferator Activated Receptor Gamma (PPAR-Gamma)	Inactive	0.99
Tox21-Stress response pathways	Nuclear factor (erythroid-derived 2)-like 2/antioxidant responsive element (nrf2/ARE)	Inactive	0.88
Tox21-Stress response pathways	Heat shock factor response element (HSE)	Inactive	0.88
Tox21-Stress response pathways	Mitochondrial Membrane Potential (MMP)	Inactive	0.7
Tox21-Stress response pathways	Phosphoprotein (Tumor Supressor) p53	Inactive	0.96
Tox21-Stress response pathways	ATPase family AAA domain-containing protein 5 (ATAD5)	Inactive	0.99
Molecular Initiating Events	Thyroid hormone receptor alpha (THRα)	Inactive	0.9
Molecular Initiating Events	Thyroid hormone receptor beta (THRβ)	Inactive	0.78
Molecular Initiating Events	Transtyretrin (TTR)	Inactive	0.97
Molecular Initiating Events	Ryanodine receptor (RYR)	Inactive	0.98
Molecular Initiating Events	GABA receptor (GABAR)	Inactive	0.96
Molecular Initiating Events	Glutamate N-methyl-D-aspartate receptor (NMDAR)	Inactive	0.92
Molecular Initiating Events	alpha-amino-3-hydroxy-5-methyl-4-isoxazolepropionate receptor (AMPAR)	Inactive	0.97
Molecular Initiating Events	Kainate receptor (KAR)	Inactive	0.99
Molecular Initiating Events	Achetylcholinesterase (AChE)	Active	0.69
Molecular Initiating Events	Constitutive androstane receptor (CAR)	Inactive	0.98
Molecular Initiating Events	Pregnane X receptor (PXR)	Inactive	0.92
Molecular Initiating Events	NADH-quinone oxidoreductase (NADHOX)	Inactive	0.97
Molecular Initiating Events	Voltage gated sodium channel (VGSC)	Inactive	0.95
Molecular Initiating Events	Na+/I- symporter (NIS)	Inactive	0.98
Metabolism	Cytochrome CYP1A2	Inactive	0.76
Metabolism	Cytochrome CYP2C19	Inactive	0.87
Metabolism	Cytochrome CYP2C9	Active	0.56
Metabolism	Cytochrome CYP2D6	Inactive	0.63
Metabolism	Cytochrome CYP3A4	Active	0.71
Metabolism	Cytochrome CYP2E1	Inactive	0.98

**Table 4 life-15-01828-t004:** Molecular Docking Analysis of Medicarpin Against Ten Selected Cancer-Related Proteins.

No.	Protein Name	PDB	Compound and Positive Control	Binding Energies (kcal/mol)	Inhibition Constant (nM)
1	CASP3	1RE1	Medicarpin	−6.98	7.71 uM
			NA3501	−7.85	1.77 uM
2	ESR1	6VPF	Medicarpin	−7.65	2.47 uM
			53Q	−3.64	2.15 mM
3	SRC	6WIW	Medicarpin	−7.08	6.51 uM
			I14	−6	40.11 uM
4	CCND1	9CSK	Medicarpin	−8.06	1.24 uM
			Sancycline	−8.1	1.16 uM
5	MTOR	5OQ4	Medicarpin	−9.6	1.57 uM
			A3W	−7.53	3.0 uM
6	PIK3CA	7R9V	Medicarpin	−6.39	20.65 uM
			2Q7	−11.31	5.11 nM
7	PARP1	7KK4	Medicarpin	−7.32	4.33 uM
			09L	−11.5	3.73 nM
8	GSK3B	4PTE	Medicarpin	−6.8	10.45 uM
			2WF	−7.2	5.26 uM
9	KDR	3VHE	Medicarpin	−7.07	6.63 uM
			42Q	−11.34	4.91 nM
10	KIT	4U0I	Medicarpin	−7.35	4.09 uM
			0LI	−13.29	180.87 pM

**Table 5 life-15-01828-t005:** Interaction Analysis of Medicarpin Against mTOR Proteins.

**Medicarpin–mTOR complex**	**Hydrogen Bonds**						
	**Docking**	**MD Simulation Timeline**					
		**50 ns**	**100 ns**	**150 ns**	**200 ns**	**250 ns**	**300 ns**
	VAL882, THR887 and LYS890	ALA805, ILE881, and THR887	ALA805, SER806, VAL882 and THR887	ALA805, SER806, VAL882 and THR887	ALA805, SER806, VAL882, THR887, ASP950 and GLY970	ALA805, SER806, VAL882, THR887 and ASP950	ALA805, SER806, VAL882 THR887 and ASP950
**Medicarpin–mTOR complex**	**Hydrophobic interactions**						
	**Docking**	**MD simulation timeline**					
		**50 ns**	**100 ns**	**150 ns**	**200 ns**	**250 ns**	**300 ns**
	ILE831, TYR867, ILE881, MET953, PHE961 and ILE963	MET804, TRP812, ILE831, TYR867, ILE879, ILE881, MET953 and ILE963	MET804, TRP802, ILE831, TYR867, ILE879, ILE881, MET953 and ILE963	MET804, TRP812, ILE831, TYR867, ILE881, MET953 and ILE963	MET804, TRP812, ILE831, TYR867, ILE879, ILE881, MET953, ILE963 and ILE968	MET804, TRP812, ILE831, TYR867, ILE879, ILE881, MET953, ILE963 and ILE968	MET804, TRP812, ILE831, TYR867, ILE879, ILE881, MET953, PHE961, ILE963 and ILE968

## Data Availability

The datasets generated and/or analysed during the current study are available from the corresponding author on reasonable request. Public third-party resources used are accessible at: GeneCards (https://www.genecards.org/), SwissADME (https://www.swissadme.ch/), SwissTargetPrediction (https://www.swisstargetprediction.ch/), SEA (http://sea.bkslab.org/), ProTox-III (https://tox-new.charite.de/protox_III/), and KEGG (https://www.kegg.jp/). Re-use of KEGG pathway imagery is covered by permission from Kanehisa Laboratories (permission letter supplied).

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
