# Peer review of "Computational Investigation Identifies mTOR as a Primary Binding Target of Medicarpin in Cholangiocarcinoma: Insights from Network Pharmacology and Molecular Docking"

_life, 2025, doi:10.3390/life15121828_

Round 1
Reviewer 1 Report
Comments and Suggestions for Authors
This manuscript presents an in-silico study integrating target prediction, network pharmacology, KEGG/GO analysis, docking, and MD simulation to explore the molecular mechanism of medicarpin in CCA. Overall, the topic is interesting and relevant, as cholangiocarcinoma remains a challenging malignancy with poor therapeutic options, and natural-product–based drug discovery is an important research direction. The study is logically structured, includes comprehensive ADME/Tox assessment, and presents extensive docking/MD analysis indicating that mTOR may be a key target. The topic is within the scope of the “Pharmaceutical” section of Life and the iThenticate score is OK.
However, the manuscript has several weaknesses that must be addressed before being considered for publication.
Detailed comments
Line 41, The claim that mTOR is “identified as a pivotal mechanistic hub” is too strong without wet-lab validation. Please rewrite.
Lines 58-59, relevant references should be added: https://doi.org/10.56782/pps.142 and https://doi.org/10.56782/pps.147
It is not clear how many compounds were screened? Only medicarpin?
There is inconsistency: first, docking performed with AutoDock Vina; later with AutoDock 4.2. Please clarify why two separate protocols were used and how results were reconciled.
A survival analysis based on n=18 vs 18 patients is severely underpowered and should not be overinterpreted. Kaplan–Meier outcomes should be presented as exploratory only.
The ADME section is disproportionately long and contains excessive tabular data. I suggest to move part of this section to SI.
Why the Authors have used the Schrodinger Maestro for MD while AutoDock for docking and not, i.e. Glide, which is a part of Maestro? Choosing one software would increase the stability of the results, as it would allow to use the same forcefield and other simulation setting. This should be explained.
The Discussion heavily reiterates the Results rather than interpreting them. Please enhance critical reflection, particularly regarding limitations such as:
– no experimental validation
– single ligand focus
– predictive target-screening biases
– inconsistent survival trends
Reviewer 2 Report
Comments and Suggestions for Authors
Dear Authors,
Introduction: This part is very brief and lacks some information about known inhibitors.
In the material and methods, and results
1) Miss the PDBID of the protein used for the study. It is important to comprehend the study (active site, interactions, etc)
2) How many MD replicas have you conducted?
3) Improve the MD analysis. Explain whether the interactions observed during the docking study were maintained or changed, and how this relates to the trend of the MD. Add a table showing this.
See [https://doi.org/10.3390/ijms24119609] for more details.
Round 2
Reviewer 1 Report
Comments and Suggestions for Authors
The authors have revised and improved their work, current version can be accepted.
Reviewer 2 Report
Comments and Suggestions for Authors
Dear Authors,
The paper is ready for publication.